# Effectiveness of Workplace Nutrition Programs on Anemia Status among Female Readymade Garment Workers in Bangladesh: A Program Evaluation

**DOI:** 10.3390/nu11061259

**Published:** 2019-06-03

**Authors:** Muttaquina Hossain, Ziaul Islam, Sabiha Sultana, Ahmed Shafiqur Rahman, Christine Hotz, Md. Ahshanul Haque, Christina Nyhus Dhillon, Rudaba Khondker, Lynnette M. Neufeld, Tahmeed Ahmed

**Affiliations:** 1Nutrition and Clinical Services Division, icddr,b, Dhaka 1212, Bangladesh; zia@icddrb.org (Z.I.); tnaherster@gmail.com (A.S.R.); ahshanul.haque@icddrb.org (M.A.H.); tahmeed@icddrb.org (T.A.); 2Global Alliance for Improved Nutrition (GAIN), 1202 Geneva, Switzerland; ssultana@gainhealth.org (S.S.); christinehotz.to@gmail.com (C.H.); cnyhus@gainhealth.org (C.N.D.); rkhondker@gainhealth.org (R.K.); lneufeld@gainhealth.org (L.M.N.); 3James P. Grant School of Public Health, BRAC University, Dhaka 1212, Bangladesh; 4Department of Global Health, University of Washington, Seattle, Washington, DC 98104, USA

**Keywords:** anemia, readymade garments, female workers, workplace program, Bangladesh

## Abstract

Eight in ten female readymade garment (RMG) workers in Bangladesh suffer from anemia, a condition which damages both health and productivity. This study evaluated the effectiveness of a workplace nutrition program on anemia reduction in female RMG workers of Bangladesh. A quasi-experimental mixed method study was conducted on 1310 non-pregnant female RMG workers from four factories. Two types of intervention packages (A and C) were tested against their respective controls (B and D) over a 10-month period. Among factories that already provided lunch to workers with regular behavior change counseling (BCC), one intervention (A) and one control (B) factory were selected, and among factories that did not provide lunches to their workers but provided regular BCC, one intervention (C) and one control (D) factory were selected: (A) Lunch meal intervention package: daily nutritionally-enhanced (with fortified rice) hot lunch, once weekly iron-folic acid (IFA) supplement and monthly enhanced (with nutrition module) behavior change counseling (BCC) versus (B) Lunch meal control package: regular lunch and BCC; and (C) Non-meal intervention package: twice-weekly IFA and enhanced BCC versus (D) Non-meal control package: BCC alone. Body weight and capillary hemoglobin were measured. Changes in anemia prevalence were estimated by difference-in-difference (DID) method. Thematic analysis of qualitative in-depth interviews with RMG workers was performed and findings were triangulated. Anemia was reduced significantly in both lunch meal and non-meal intervention (A and C) group (DID: 32 and 12 percentage points, *p*: <0.001 and <0.05 respectively). The mean hemoglobin concentration also significantly increased by 1 gm/dL and 0.4 gm/dL in both A and C group (*p*: <0.001 respectively). Weight did not change in the intervention groups (A and C) but significantly increased by more than 1.5 kg in the comparison groups (B and D). The knowledge of different vitamin and mineral containing foods and their benefits was increased significantly among all participants. Workplace nutrition programs can reduce anemia in female RMG workers, with the greatest benefits observed when both nutritionally enhanced lunches and IFA supplements are provided.

## 1. Introduction

The readymade garment (RMG) industry is one of Bangladesh’s most important industries and contributes 81% of total export earnings and more than 14% of the GDP (2016–2017) [1]. The sector directly employs approximately 3 million women [1]. Although this industry has contributed to the growth of the economy and foreign earnings, it faces many limitations with respect to the low wage rate, insecure employment conditions, as well as the lack of safety and respect for female workers’ rights. Due to a low minimum wage, garment workers cannot afford adequate and appropriate food as well as clothing, housing, transportation cost, and medical expenditures [2]. A recent study demonstrated that 77–80% of female Bangladeshi RMG workers are anemic [3]. Anemia is associated with a significant productivity loss in terms of GDP [4], increased maternal and perinatal mortality, and contributes to global mortality [5]. Anemia is primarily caused by iron deficiency but also by other micronutrient deficiencies such as vitamins A, B2, folate, B12, and minerals like selenium, and copper [6], and as well as several other non-nutritional causes [7]. In Southeast Asia, it has been estimated that iron deficiency anemia is the cause of 26% of all anemia in women of reproductive age [8]. However, there is a scarcity of evidence on the etiology of anemia and their relative proportion to the causes of anemia. 

The inclusion of micronutrient-rich foods in the daily diet, such as red meat, green leafy vegetables and some nuts/seeds, is often not affordable for populations living under conditions of poverty in countries like Bangladesh, but may be addressed at the workplace if meals are provided for workers. In this context, food fortification could also be a cost-effective contribution to food-based approaches for preventing anemia by increasing the intake of iron and other micronutrients, a lack of which can cause anemia. The fortification of staple foods is advantageous because it does not require the population to change their dietary habits and allows fortification with multiple micronutrients since deficiencies often occur concurrently [9]. Studies from Latin America, Africa, and India showed that rice fortification is safe and effective in improving micronutrient status, with the impact commensurate with the micronutrient content of the fortified rice, but as much as 53–80% [10,11,12,13]. In rice-consuming countries like Bangladesh, multiple micronutrients fortified rice could be a promising strategy to address micronutrient deficiencies and thus reduce anemia. The workplace where RMG workers spend most of their waking hours could be an important entry point to address this substantial burden of anemia. The primary means of addressing anemia in this population of female garment workers in the short term will be through dietary modifications and rice fortification, complemented by iron/folate supplements (IFA). The benefits would be further if behavior change communication (BCC) messages are combined. BCC not only improves women’s dietary practices [14] but also improves the compliance in IFA intake [15], which further helps to reduce the risk of anemia [16]. The non-nutritional causes of anemia, i.e., poor hygiene leading to infection, inflammation due to multiple causes including malaria, helminths, and chronic infections like tuberculosis [17] could also potentially be addressed through BCC. 

In this study, a workplace nutrition program was implemented in four RMG factories in Bangladesh to reduce anemia of female workers. The workplace program included a package of interventions which included nutritionally-improved lunches and weekly IFA supplements OR twice weekly iron supplements, as well as BCC (modules on nutrition including balanced diets/dietary diversity, iron/folate-rich foods, infant and young child nutrition (IYCN), anemia prevention, hand-washing with soap and dietary diversity), over a period of 10 months. The guidelines for nutritionally improved lunches available in factory canteens included: green leafy vegetables, lentils, fortified rice, fortified oil, and iodized salt every day and flesh foods at least three times a week. Regular lunches included non-fortified rice, lentils and one portion of vegetable daily and meat or fish or egg three times weekly, cooked with fortified oil and iodized salt. In these factories, IFA was provided once weekly, due to the additional iron supplied through fortified rice. 

The evaluation of the effectiveness of such a nutritional intervention package is essential before scaling up. Hence, the current study aimed to evaluate the use of a different combination of packages to reduce anemia among female garment workers in these selected garment factories in Bangladesh. The results of this evaluation would help to inform government policymakers, donors, factory owners, and female garment workers about the benefits of the nutrition intervention and to advocate for the inclusion of appropriate nutrition interventions packages in the workplace in garment factories of Bangladesh. 

## 2. Materials and Methods 

### 2.1. Study Design and Study Site

A quasi-experimental, mixed method study design was applied in four RMG factories for the evaluation of the different packages of intervention to measure their impact on anemia reduction (Figure 1). The baseline survey was conducted between December 2015 and February 2016 and the endline was conducted between January and March 2017. Each factory employs about 4000–6000 workers the majority of which are young women. The factories operate on six workdays per week. A total of 1310 participants from four factories participated at baseline and 1290 participants were followed up from all four factories at endline. Two types of intervention packages were tested against two corresponding comparisons with a minimal package. Among factories that already provide lunch to workers, one intervention (A) and one comparison (B) factory were selected, and among factories that do not provide lunches to their workers, one intervention (C) and one comparison (D) factory were selected. The lunch combination was chosen to find out whether (A) fortified lunch combined with IFA is more effective in reducing anemia compared with (B) regular lunch with non-fortified rice and twice-weekly IFA. This package would be helpful for the factories that were already providing or plans to provide a lunch meal to their workers. On the other hand, there were factories which did not provide lunch meal to their workers. So, we have chosen an intervention package (C) and control package (D) for non-lunch providing factories to find out the effective package for anemia reduction. The factories were selected based on the factory owner’s willingness to participate in this study. 

Factories A, C, and D were situated in the Bangladesh Export Processing Zone in Savar Upazila of Dhaka district. Factory B was situated in an industrial area of Bhaluka Upazila of Mymensingh district. 

### 2.2. Program/Intervention Description

The lunch intervention consisted of a nutritionally enhanced lunch meal with multi-nutrient fortified rice, weekly iron-folic acid (IFA) supplement to female workers, and a behavior change communication (BCC) program conducted in the factory setting and the comparison factory participants received regular lunch with non-fortified rice and BCC. The fortified rice is composed of a local variety of rice blended with a multi-nutrient fortified extruded rice product pre-mix which contains 150 µg vitamin A RE, 0.4 mg thiamin, 1 µg vitamin B12, 130 µg folic acid, 6 mg iron (Ferric pyrophosphate), and 4 mg zinc per 100 g. The IFA tablet contained Ferrous fumarate 200 mg and folic acid 400 µg. The non-lunch intervention participants received twice-weekly IFA and once monthly BCC and the comparison factory participants received BCC only. Additional modules were added to the existing health-focused BCC activities (Table 1). The modules were on balanced diets/dietary diversity, iron/folate-rich foods, infant and young child nutrition (IYCN), and an overview module on anemia prevention, hand washing with soap and dietary diversity.

#### Composition of the Different Packages with Rationale

Lunch meal: During a situation analysis of seven factories prior to this study, it was found that the regular factory lunch only provides 761 kcal of energy (Appendix A) out of recommended 2200 kcal of energy [18]. To reduce the gap in energy contents, the enhanced lunch menu was designed to include the following changes: (i) an increase in the thickness of lentils by increasing the lentil content per cooked volume by a factor of two; (ii) increasing the portion sizes of meat/fish/poultry while continuing to serve at least three days per week; (iii) increasing the portion size of egg while continuing to serve at least 1 day per week; (iv) increasing the serving size and frequency of serving mixed vegetables to 6 days per week, and; (v) including a serving of green leafy vegetables six days per week. The frequency and portion sizes for regular and fortified lunch menu changes are summarized in Appendix A. These changes were derived to conform more closely to the Bangladesh Dietary Guidelines [18] for intakes of macronutrients and food groups. The regular and fortified lunch menu was compared to the suggested portion sizes for different food groups outlined in the Guidelines, using the assumption that the lunch meal would provide 40% of the total day’s dietary intakes (Appendix A). While the regular menu meets only one criterion other than calorie intake. The energy and nutrient content of the fortified lunch meals provided in factories were estimated based on information provided by food service providers in factories that participated in the Situation Analysis (Appendix A). The percentage contribution of the lunch meal to nutrient requirements was calculated using the WHO/FAO recommended nutrient intakes (RNIs) for non-pregnant, non-lactating women of reproductive age [19]; protein and fat content were expressed as the percent contribution to total energy intakes as per the Bangladesh Dietary Guidelines [18] (i.e., 20% for protein and no more than 30% for fat). The energy and nutrient content of the diet was then recalculated after substituting non-fortified rice for Ultra Rice and non-fortified vegetable oil with vitamin A-fortified vegetable oil. Based on these estimates, the contribution of the intervention diet to iron, zinc, folate, vitamin B12, and vitamin A was increased substantially.

Iron/folate supplement: To increase the intake of iron and folate, an intermittent iron/folate supplement was provided to all women of reproductive age with a view that the lunch provided by the factory was not sufficient to meet the daily iron-folate demand (Appendix A). The distribution of iron/folate supplements to women at risk of anemia in the context of the workplace is consistent with current local recommendations for use of alternative mechanisms prevent and control anemia among populations with low utilization of health services [20] and international recommendations for menstruating women in populations with prevalence of anemia >20% [21]. The dosing regime was as follows:
(i)Intervention factory with a lunch program: the iron/folate supplement was provided once weekly to non-pregnant women of reproductive age. This increased the mean daily intake equivalent of iron by 8.6 mg/day and of folate by 57 µg/day, equivalent to 15% or 29% of the RNI for iron at low or moderate bioavailability, respectively, and to 14% of the RNI for folate. (ii)Intervention factory without a lunch program: the iron/folate supplement was provided twice weekly to non-pregnant women of reproductive age. This will increase the mean daily intake equivalent of iron by 17.1 mg/day and of folate by 114 µg/day, equivalent to 30% or 58% of the RNI for iron and low or moderate bioavailability, respectively, and to 29% of the RNI for folate. The twice-weekly doses considered, as they were not getting fortified rice so there will be a gap in iron-folate requirements (Appendix A).

Behavior Change Communications (BCC) Component: The role of BCC was well established for dietary modification along with better compliance to IFA intake and therefore reduction in the risk of anemia. So, it was provided to all factory workers. The modules of BCC and enhanced BCC were mentioned in Table 1. Enhanced BCC included nutrition module, which was not added on the regular one. Additionally, there was a summary module with the key messages at the beginning of the intervention to provide awareness on hand washing, anemia prevention, iron/folic acid supplements, and dietary diversity. Even the worker’s, who received neither lunch nor IFA, received BCC. BCC sessions were provided to improve the factory worker’s knowledge on the importance of the key messages delivered and practicing behavioral change at an early stage of the program. The messages were then be reinforced throughout the program to aid in the retention of information. The process for the retention of knowledge through BCC is vital in the context of Bangladesh where anemia is quite prevalent among RMG workers. 

### 2.3. Sample Size

The sample size for the study was calculated based on the ability to detect a 15-percentage point reduction in the prevalence of anemia from the baseline because of the impact of lunch interventions packages (A or C) compared to the comparison (B or D) [11]. Each factory served as an individual cluster. Considering a 5% level of significance, 80% power, 1.5 design effect (DEFF) to adjust for intra-class correlation (ICC), and 20% loss to follow up, the minimum sample-size was 328 female garment workers required per factory. At baseline, we enrolled 328 participants from each of Factories B, C and D and 326 from Factory A, for a total of 1310 participants. At endline, 1290 participants were followed up from all four factories. Among them, 328 participants were followed up each from (B), (C) and (D), and 306 participants from (A).

### 2.4. Enrolment of Study Participants 

Before starting the study, we sought permission to work in the factories. At the start, the factory management and superintendents were informed in detail about the objectives and procedure of the study. Subsequently, the study was explained to all factory employees through meetings. The investigators received a list of non-pregnant female garment workers from the factory human resource management office at least three days prior to each survey. Reporting pregnancy status to the factory authority is mandatory for the workers. Each list contained the name, age, date of birth, factory identification number and pregnancy status. At first, the sample list was cleaned according to study eligibility criteria: adult non-pregnant female garment worker between 18–42 years of age from selected factories who were present and gave consent for the interview, not suffering from any known chronic illnesses. After that, the cleaned list sorted according to the five age categories (18–22 years, 23–27 years, 28–32 years, 33–37 years, and 38–42 years). Participants were then randomly selected from each age category. By this process, the study investigators made a sample list of 328 workers from the sample frame. The participants who were enrolled during baseline were supposed to be followed up again during endline. At endine, 30–40% of the subjects were unavailable (they had either left the job or were on leave during the endline survey). Therefore, we decided to replace the participants at endline with the aforementioned matching criteria. The replacement participants list was prepared based on the study eligibility criteria with an addition that the replaced participant would be within the same age category, working in a similar position for the same duration in that factory, not pregnant and had received the intervention for at least the last eight months. Sensitivity analysis between the loss to follow-up participants and replaced participants revealed that the replaced participants have similar baseline socio-demographic status and hemoglobin levels as participants lost to follow up (Appendix A). We also did an analysis without the replaced participants and found similar results, which make us confident of using this replacement approach (Appendix A).

### 2.5. Questionnaires

Trained research assistants applied a semi-structured questionnaire to collect data on background information and the socio-economic status of study participants and their respective households, food recall for the last seven days, water, sanitation, and hygiene practices, morbidity in the last 14 days and sick leave in the last 30 days preceding the interview. The questionnaire was pre-tested before the baseline survey. 

### 2.6. Anthropometric Measurement

Weight was measured in kilograms (kg) using a portable Tanita scale. The participants were weighed in regular clothing. The scale was placed on a flat surface, and the participants were asked to stand still on the scale so that the scale would display a weight. One research assistant read the weight and the other wrote it on the form. The participants were weighed to the nearest 100 g. The participants were weighed twice, and the mean was used for further analysis. All devices and measurement procedures were re-calibrated each day prior to data collection.

### 2.7. Blood Sample Collection and Analysis

Trained research staff collected capillary blood from all participants through pricking the fingertips by a lancet, while a strict aseptic environment was maintained throughout the process. A separate lancet was used for each individual and hemoglobin concentration was determined using microcuvettes and six Hemocue portable hemoglobinometers (HemoCue HB301 analyzer, HemoCue AB, Angelhom, Sweden) at baseline and endline inside the factory. Hemoglobin determinations made using this equipment were validated previously in a Bangladeshi population [22]. The research staffs were trained on the blood collection procedure and aseptic techniques by a researcher before both surveys. Researchers frequently visited the field and rechecked 2% of participants for their Hb concentration to validate the enumerators’ activities. The cut-off levels of hemoglobin concentration to define anemia were <120 g/L for non-pregnant non-lactating or lactating women as the survey areas are at sea level [23]. The same HemoCue and weight machines that were used in baseline were used again during the end line survey to ensure measurement reliability and validity. Two research staff checked and calibrated the machines each morning before starting the measurements. All data from each participant was collected on a single day. The baseline and endline surveys were conducted in the same season.

### 2.8. Qualitative Interviews

The qualitative approach was planned *a priori* to understand—(a) the regular food intake both at factory and home, during weekdays and weekends, food preferences and opinions on factory lunch, (b) knowledge and practice of nutrition-related topics included in the BCC training, (c) attitudes, intakes, and experiences with IFA supplements. We also evaluated the possibility of confounding, i.e., if something external happened to the intervention factories and their communities to decrease anemia, e.g., the presence of a community screening program, presence of other programs/intervention, IFA or any type of vitamin/supplements intake by the participants outside of the factory. The interviewers selected eight participants from each factory by purposeful sampling [24]. These participants included 5–6 RMG workers, one ‘*Shasthoshokhi*’ (peer educator) and one severely anemic participant identified during the baseline survey from Factory A and C (Factory B and D had no severely anemic participant at baseline). Different types of participants were selected to understand the variation of responses (if any). The participants were selected if they were working in that particular factory for at least 10 months, had participated in the baseline survey and were willing to participate at their home during the weekend. By this process, 32 in-depth interviews (IDIs) were conducted from four factories which took place at participants’ homes on holidays at their convenient time. The participant’s home setting was chosen to avoid workplace related influences on their response.

### 2.9. Outcome

The primary outcome was the difference in anemia at the end of 10 months from the intervention compared with the respective comparison arms. In addition, the changes in hemoglobin concentration and the existence and severity of anemia from baseline to endline were assessed. The study also assessed several secondary outcomes including (i) difference in knowledge on health and nutrition delivered to female RMG workers from baseline to endline, (ii) difference in self-reported morbidity in the last 14 days preceding the interview from baseline to endline intervention.

### 2.10. Data Management and Statistical Analysis 

#### 2.10.1. Quantitative Data

A trained data manager using Microsoft Access 2007 performed data entry and validation of entry of questionnaires and anthropometry forms. Data management and statistical analyses were executed using SPSS software (version 20, SPSS Inc., Chicago, IL, USA) [25]. The summary statistics were expressed as means with standard deviations, medians or percentages, with 95% confidence intervals (CI). The response rate of the participants was 100%. To assess differences in mean levels among the factories, the student *t*-test, and for categorical outcomes chi-square statistical comparisons of proportions with 95% confidence intervals were calculated. Since the study was quasi-experimental, the impact of the intervention was assessed using difference-in-difference analysis. All independent variables were analyzed initially in bivariate models and the attributes that were significantly associated with anemia (dependent variable) and biologically plausible were included in the linear mixed effect regression models with time-group interaction and baseline anemia levels. Since each factory was considered as a single cluster, the intracluster correlation coefficient (ICC) was also adjusted in the model. The model was further adjusted by including the differing baseline characteristics (Appendix A) to find out the true effect of the intervention. *P*-values less than 0.05 were considered significant. 

#### 2.10.2. Qualitative Data

The qualitative checklist was developed by study investigators a priori during the development of the protocol with pre-coded themes to avoid discrepancies among and between interviews. IDIs were written in detail and transcribed verbatim into Bangla, and then translated back into English by a bilingual speaker. Participants were noted in transcripts as P1, P2, etc. in order to link statements with demographic data collected during the quantitative interview. English transcripts were analyzed manually and double-coded by two trained qualitative researchers. The coding team collectively developed the codebook based on the interview checklist and from emergent findings. The agreement was achieved in coding through discussion on codes after initial coding; the study team arbitrated any disagreements. Direct quotes were extracted from interviews and linked to the findings from quantitative interviews. Results were analyzed according to four themes: (a) workers lunch experience and satisfaction, (b) knowledge, opinion and practice on BCC topics, (c) knowledge and experience on both provision and consumption of IFA tablet, (d) experience on illness and absenteeism during the last 30 days. The response rate of the participants was 100%. The findings from the qualitative interviews were triangulated with quantitative findings. 

### 2.11. Ethical Considerations

The study was approved by the Institutional Review Board of icddr,b. Written informed consent was collected from all study participants prior to enrolment by signature. The ethics committees approved the consent format prior to data collection. The researchers sought permission from the factory owner to conduct interviews on their workers inside the factory setting in a separate room to ensure their privacy and confidentiality. The study investigators had to share the list of the participants with the factory authority prior to the interview so that their supervisors would allow them to participate, which might have introduced some biased responses. The data collectors tried to avoid such bias by better rapport building and motivating the participant to share their true information. The data collectors strictly maintained the code of ethics for conducting the interviews and reported no breach of confidentiality during the interview conduct. Hemoglobin status was revealed only to the participants with severe anemia who were advised to seek necessary medical attention. However, the study team did not provide any medical service to any participant and could not ensure the medical care needed for the severely anemic participant. This trial was registered at the clinicaltrials.gov as NCT03073590.

## 3. Results

### 3.1. Baseline Socio-Demographic Status

At baseline, 1310 female garments workers participated in the survey, the majority were aged between 23–32 years, Muslim and married (Table 2). However, more participants were unmarried in A and C compared to their respective comparison factories (*p* < 0.004 and 0.005 respectively). A higher proportion of participants from Factory A lived in rented houses whereas almost half of the participants from Factory B lived in their own house. The asset quintile significantly differed among lunch meal factories. Almost one-third of the participants from factory A were from the richest asset quintile compared to B, where most belonged to the lowest asset quintile. More than 94 percent of participants worked overtime with an average of 40 overtime hours per month. Significantly, more participants worked overtime in factories B and D compared to their peers. Therefore, they had more income and expenditure at baseline within the same factory. The median income of these participants varied between the equivalents of 100 to 109 USD per month. Total median expenditure varied between 94 to 106 USD per month. Participants from factory A and C were significantly more anemic at baseline compared to their respective controls. The response rate of the participants was 100% at both baseline and endline surveys.

### 3.2. Baseline Factory Food Experience and Satisfaction

At baseline, most of the workers from all four factories took three major meals (breakfast, lunch, and dinner) each day. They occasionally skipped meals. Workers from lunch meal factories mostly took lunch at factories and non-lunch factory workers went home to consume their lunch. The majority of workers from lunch meal providing factories were very satisfied with their lunch experience (more than 90% workers). The workers from lunch meal factories mentioned that the lunch is good in quality, quantity, and taste and it satisfied their hunger. We also asked all participants about their regular food intake at home during breakfast and dinner and found that almost all of the workers take rice, fish, and vegetables at home. Although workers from the factory A and factory B quite often consume locally available inexpensive fish, i.e., tilapia and pangash, 3–4 times a week, workers in the non-lunch group (factory C and D) mostly consumed rice with vegetables. 

### 3.3. Food and Nutrition Knowledge and Adherence to Iron-Folic Acid (IFA) Tablet

During the IDIs, the majority of workers from the factories A, B and C reported consistent BCC sessions in the 6 months preceding the endline survey in comparisons with factory D were the sessions were inconsistent. The inconsistency in this particular factory was due to the factory management’s lack of support for the program. The training was quite irregular and short but was very informative and useful as reported by the workers. From baseline to endline quantitative surveys, more than half of the participants heard about any of the main food groups and this change was statistically highly significant (Table 3). A knowledge of different vitamin and minerals, vitamin A and iron-containing foods and their benefits significantly increased among participants in all factories from baseline. At endline, about 60% of participants mentioned green leafy vegetables as a source of vitamin A. Participants were well aware of the benefits of vitamin A (~95%) and mentioned that vitamin A is good for health (77% to 89%) and eyesight (32% to 55%). Almost half of the study participants knew about the availability of vitamin A fortified oil in the market, which is an almost 2-fold increase from the baseline survey. Each of the indicators was calculated based on a single question response asked during the quantitative surveys. The overall response rate was 100%. 

Despite the significant improvement in health and nutrition knowledge, very few workers from factory A knew about the benefits of IFA tablets as reported during IDI’s. As a result, factory A respondents reported low adherence to iron-folic acid (IFA) during IDI’s. Only 3 out of 8 participants reported regular intake of IFA. Those who took the tablets regularly were quite satisfied with their health effects and some of them even mentioned that “these tablets do not harm instead it does well. My tiredness decreased”. Another worker added, “It reduces deficiencies”. The suggestions also came from the workers that “it would be better if counselors could discuss the benefits of iron tablets and what are the risks of not taking it. Everyone will take it then”. The scenario is the opposite among workers from factory C, where all eight participants reported regular intake for the last six months and no side effects from IFA. A single BCC session on the topic of anemia prevention and management was conducted in both intervention factories (A & C) seven months preceding the endline survey. However, the duration of the BCC session and coverage varied among the factories and even within the groups between the factories. We also asked the participants about the presence of any community anemia-screening program, community-based nutrition program/intervention, IFA or any type of vitamin/supplements intake by the participants outside of the factory. However, none of the participants reported on the presence of any such programs at the community. 

### 3.4. Water, Sanitation and Personal Hygiene Practice

Almost all participants from all four factories confirmed the availability of clean and safe water for drinking at the workplace in both baseline and endline surveys (92% to 99%), which is mandated by the law. Hand washing practices after defecation significantly improved only in the non-lunch comparison factory. The reported use of sanitary napkins during menstruation significantly increased in factory A, C, and D (*p* < 0.001) (Table 4).

### 3.5. Self-Reported Sickness and Absenteeism

As reported by the workers during the surveys, the prevalence of common illnesses like diarrhea, dysentery, fever, common cold, urinary tract infection, joint pain significantly reduced in Factory A from baseline to endline. The reported prevalence of these illnesses, however, increased in the factory C and D from baseline (Table 4). Absenteeism due to sickness increased significantly from baseline in Factory D, although the percentage is quite low. The median duration of leave due to sickness was increased by 1 day among all workers from baseline to endline. The IDIs revealed that lunch meal factories (A and B) easily permit their workers to take leave during sickness. The management did not deduct salary or benefits for sick leave if a proper reason was provided. Several participants mentioned, “Salary is not cut if the leave is approved. If we take leave personally then salary and attendance benefits are cut” (Factory A, age 29). Even sometimes, the factories (A and B) bear the cost of medicines and treatment. On the contrary, very few participants from factories C and D take leave because the factory authorities seldom approve leave for either sickness or any personal reason, and most often sick days were deducted from their salary. Therefore, the workers from factory D overall take very few leave days, whether personal or due to sickness. One Factory D worker mentioned having taken only four days to leave from work over three years of service, and all four days were deducted from her salary. Most of the participants showed interest in learning about the management of common illnesses (common cold, fever, diarrhea, etc), and workplace exercises for joint pain during the IDIs.

### 3.6. Change in Hemoglobin (gm/dL) and Weight (kg) Over Time 

From baseline to endline, the mean hemoglobin of female workers significantly improved by 0.7gm/dl in Factory A (*p* < 0.001) but significantly reduced by 0.3 gm/dL and 0.22 gm/dL in Factory B and D (*p* = 0.001 and *p* = 0.04 respectively). Figure 2 shows the distribution of Hb at baseline and follow-up in each group.

Overall, the weight did not change in Factory A or C, but significantly increased by more than 1.5 kg in the comparison factories (B) and (D) (Table 5). 

### 3.7. Changes in Anemia Prevalence, Hemoglobin Concentration and Anemia Severity over Time

Table 6 shows that anemia prevalence was significantly reduced by 23 percentage points in the lunch meal intervention factories. Using the DID analysis, against the relevant comparison groups, there was a significant reduction in anemia prevalence in both intervention groups, whether a meal was provided or not. Anemia reduced significantly by 32 percentage points in the lunch meal intervention after adjusting for baseline differing characteristics such as marital status, asset index, household ownership, education, and overtime work hours per month and intracluster correlation (ICC). The covariate-adjusted model showed an anemia reduction of 12 percentage points in the non-lunch intervention factory (Table 6). Mean hemoglobin (Hb) concentration was significantly increased by 1 gm/dL in the lunch meal intervention factories (Table 6). Using the DID analysis, against the relevant comparison groups, there was a significant increase in Hb concentrations in both intervention groups, whether a meal was provided or not. Hb increased significantly by 1 gm/dL in the lunch meal intervention after adjusting for baseline differing characteristics such as marital status, asset index, household ownership, education, and overtime work hours per month and intracluster correlation (ICC). The covariate-adjusted model showed a 0.40 gm/dL increase in Hb concentration in the non-lunch intervention factory. Figure 2 shows that the distribution of Hb is normal and quite similar in each of the four factories at both baseline and endline. 

Percentages of severe anemia reduced significantly (*p* < 0.001) in A and C factory, remain unchanged in B and significantly increased in D (*p* < 0.001) from baseline. Overall, the proportions of participants with all three types of anemia severity reduced in Factory A only. All three types of anemia severity increased in both comparison factories (B & D) from baseline (Figure 3).

## 4. Discussion

To the best of our knowledge, this is the first nutrition intervention study in Bangladesh conducted among female RMG workers, which showed the effectiveness of a combination of interventions to reduce anemia. In our study, anemia prevalence in female RMG workers was reduced by 32 percentage points through the provision of a workplace nutrition program that included a nutritionally enhanced lunch through dietary diversification which involved micronutrient-fortified rice along with weekly iron-folic acid supplements and an enhanced health and nutrition behavior change approach. In another workplace nutrition program that included twice-weekly IFA and an enhanced BCC, anemia prevalence was reduced by 12%. In the control group, where just a basic BCC approach was included in the workplace program, we saw 6% increases in anemia. It was obvious that anemia reduces as Hb concentration increases. We also found a significant increase in Hb concentration in both intervention groups, with the highest reduction of 1gm/dL among nutritionally enhanced lunch meal group. A recent Bangladeshi study also found beneficial effects of fortified rice in anemia and Hb reduction [26], but the rate of anemia reduction and an increase in Hb concentration was much lower. 

We examine what we know about the evidence for the interventions included in the package to understand what components may have driven the reduction in anemia. First, we see a mild reduction even in the ‘control’ groups were some BCC was still provided. A systematic review of 201 trials indicated that the prevalence of anemia can be reduced by 46% when fortified foods are given an over a period of 6–11 months because the hematologic response should be evident after 6 months of the intervention [27]. Our study findings also suggest that at least 10 months of intervention can reduce anemia significantly; since the prevalence of anemia is likely elevated in Bangladeshi female RMG population due to their poor dietary quality and pre-existing multiple nutrient deficiencies. A recent study in Ghana shows that micronutrient-fortified rice can be a significant source of dietary bioavailable iron [28]. Both lunch meal intervention and non-lunch intervention packages contain moderate bioavailable dosages of iron (Appendix A) but the bioavailable iron content was relatively higher among the fortified lunch meal package; due to the presence of iron in rice and iron in IFA. This also helped us to understand that despite low adherence to IFA, the anemia reduction and increase in hemoglobin concentration was significantly higher in the fortified lunch package-receiving participants compared to controls. These findings suggest that there might be the presence of iron deficiency in this population for which the intervention rich in iron worked well in anemia reduction. As we did not test iron levels in blood, the possibilities of iron deficiency anemia in this population could not be ruled out. In our study, the iron supplementation doses were within the recommended level and we did not find any health adverse outcome during the study period except for the common side effects. However, high iron levels have increased the risk of mortality and decline in cognition levels in Chinese adults [29,30]. Hence, we needed to be cautious about supplementing high doses of iron at the population level over the long-term, especially when the etiology of anemia is unknown.

The World Health Organization recommends providing intermittent doses of IFA (once or twice weekly) rather than daily doses as an effective and safer alternative to daily iron supplementation for preventing and reducing anemia at the population level, especially in areas where this condition is highly prevalent [20]. Therefore, the current study also provided intermittent doses of iron-folic acid (IFA) supplementation once or twice weekly to female RMG workers. A systematic review indicated that intermittent dosing provided to menstruating women can reduce the prevalence of anemia by 27% compared to a control, regardless of dose and frequency (i.e., 1–3 times per week), and duration of the study [31]. Our study findings also support this fact that intermittent dosing is more practical and feasible to maintain over the long term in a factory setting. 

The adherence to IFA was lower in the weekly (37%) than the twice-weekly (100%) group. Several other studies have demonstrated the impact of IFA supplementation on the reduction in the severity of anemia among non-pregnant women [31,32,33], pregnant women [34,35] and adolescent girls [36,37]. In our study, the severity of anemia significantly reduced in both lunch and non-lunch intervention factories that provide intermittent IFA along with BCC. Although, the effect of IFA with BCC on anemia reduction in our study is comparatively low, only 15 percentage points compared to the 32% reduction with IFA plus an enhanced lunch meal. This might be due to the low adherence to once weekly IFA, as most of the workers were not aware of the health benefits of IFA. They had a lack of knowledge about anemia, IFA tablet and the misconception about the side effects of the IFA tablet. In addition, BCC counseling was also irregular and short, which may have further reduced the effectiveness of IFA and BCC intervention. Even with the irregular BCC, the workers’ knowledge of nutritious food increased significantly in all four factories. In addition, the workers in three factories were more likely to report using sanitary napkins post-intervention, as was previously shown with a similar BCC program [38]. This indicates a need for greater support for the BCC program to become effective and sustainable. In a discussion with the factory higher management, the reason for not supporting the program becomes clearer. According to the management, the BCC sessions made the workers more concerned about their rights and as a result, the factory had more job turnover than ever, and it caused more harm than good to the factory benefits. We also understood in some factories that workers were not keen on the BCC because it was during their productivity time and so they were feeling constrained. Thus, BCC should be provided outside factory hours but still hours paid by management. On the contrary, empirical evidence shows that industry-based training has resulted in the most favorable outcomes in terms of productivity, worker turnover and absenteeism reduction [39]. Thus, building greater sensitivity and capacity among factory management by displaying the business benefit of BCC might help to improve the situation [40]. Estimates showed that Bangladesh loses up to 8% of its GDP due to anemia [3], which could be reversed by eliminating anemia through combined packages, including BCC. 

A recent Bangladeshi survey estimated that roughly 5% percent of anemia is caused by iron deficiency [41], although the exact amount in this population is not known. Other micronutrient deficiencies might prevail in this population, which explains the relatively lower impact of IFA on anemia reduction in our study. This finding also indicates that the significant anemia reduction might be largely due to the multi-nutrient enhanced lunch meal (21 percentage points). In addition, the workers were very satisfied with the quality, quantity, and taste of the enhanced lunch meal, which ensures the high adherence of lunch meal among the workers. The lunch meal not only reduced anemia but also was associated with an increase in the workers’ capacity of working for more hours and reduced reports of common co-morbidities like the common cold, urinary tract infection and joint pain significantly. The overall sickness absenteeism reduced in both lunch intervention and control factory, although the days of sickness absenteeism increased by one day from baseline. The IDIs revealed that the management at lunch meal factories easily permits their workers to take leave during sickness and did not deduct salary or benefits for sick leave if a proper reason was provided. This explains the relative increase in leave days due to sickness. Nevertheless, the good side was that the sick leave allowed the workers proper and rapid recovery from illness. We found a significant decrease in mean hemoglobin level and increase in anemia percentages among control factory workers from baseline. From IDIs we also identified a lack of support during sickness and family emergency and even salary cut during unauthorized leave in control factories. This might have caused long-term psychological stress and the increase in anemia from baseline [42,43]. The enhanced BCC in lunch meal workers might have helped workers to practice appropriately during their home stay, which might have stopped the continuous progression of anemia for intervention factory, which did not happen for the control group. There is clear-cut evidence that providing sick leave can reduce turnover, increase productivity, and reduce the spread of contamination in the workplace [44]. In this way, the lunch meal might have improved the work performances [45]. This could be a significant finding for this sector as improved work performance may have improved their income through fewer unpaid sick days, and possibly in increased factory productivity. Literature suggests that, eliminating anemia results in a 5–17% increase in adult productivity, which adds up to 3% of GDP in Asian countries [4,46]. Thus, an investment in nutrition in order to reduce anemia can potentially reverse productivity losses and could lead to associated benefits e.g., employee retention, reduced cost of training, reduced absenteeism, and improved employee motivation. 

### Strength and Limitation

This is the first evaluation study, which demonstrated a reduction of anemia in a population (RMG workers) that has the largest contribution to the country’s economy, but has yet never been targeted for any program or intervention. Several other studies have proven the efficacy and effectiveness of the intervention components (multiple micronutrients fortified rice, IFA, and BCC) separately [11,31]. However, this study proved the effectiveness of the combination of intervention, especially in factory settings. We also checked for the possibility of confounding, i.e., if something external happened to the intervention factories and their communities to decrease anemia, e.g., presence of community screening program, presence of another program/intervention, IFA or any type of vitamin/supplements intake by the participants outside of the factory. However, there was no such program in the community at the time of this current study. This makes us more confident about the positive effects of workplace nutrition programs on anemia reduction. The qualitative approach also helped us to understand the context, feasibility, duration, and sustainability of the intervention in factory settings and further scale-up of the intervention. The mixed method design of this study further allowed us to triangulate the study findings and to understand the different perspectives of the intervention and the resulting outcome. This project had some limitations, e.g., (a) intervention duration was only 8–10 months; (b) interventions were not systematically tracked, particularly the IFA distribution; (c) compliance of IFA was low; (d) factory non-compliance (delay in providing information of workers, interfering in scheduling and during conduction of interview time); (e) lack of intervention randomization among pilot factories might not reflect the true effect of intervention; (f) about 40% of workers were lost to follow-up from baseline, (g) replacements of participants were done only based on age category and intervention duration, and (h) possibilities of iron deficiency anemia could not be ruled out due to lack of iron level testing in blood, which could have created a spurious effect on the findings. Hence, the findings needed to be interpreted cautiously. Due to the lack of a true control group, this study could not confirm the specific effect of any single intervention (i.e., without the inclusion of a BCC). What is also not clear is whether there was any additional benefit to including weekly IFA to the intervention incorporating fortified rice. Finding comparison groups in private sector settings is very difficult, and the factories who were willing to serve as counterfactuals were not likely representative of the average garment factory. In addition, baseline data indicate the anemia prevalence in these comparison factories was substantially lower than in the intervention factories. The large baseline difference in anemia status among lunch meal groups might have had an effect on overall study impact. The impact of intervention might have been different if the comparison group also had a higher prevalence of anemia at baseline as like lunch meal intervention group. However, we have adjusted these factors in our model. The study team could not measure the food intake and changes in factory productivity as the data were collected by another organization and not available to study investigators. We also could not rule out the other causes of anemia due to hemoglobinopathies, which is not rare in our country context [47].

## 5. Conclusions

Anemia among non-pregnant female RMG factory workers can effectively be reduced by providing a combination of interventions over a significant period. Anemia reduced more significantly when the factory provided a freshly prepared, nutritionally enhanced lunch with fortified rice, increased diversity, and combined this with a weekly IFA tablet. Among non-lunch factories, IFA was able to reduce anemia significantly, but the rate of reduction was not as high as in lunch meal factories. In order to achieve the goal of a substantial reduction in the prevalence of anemia, a multidisciplinary approach is essential, with the active collaboration of all sectors involved, including government, donor agencies, local academic institutions, non-governmental organizations, and local communities. Therefore, government, policymakers and readymade garment factory owners should take initiative to implement and scale up this combined program to all garment factories in Bangladesh in order to successfully reduce and prevent anemia. 

## Figures and Tables

**Figure 1 nutrients-11-01259-f001:**
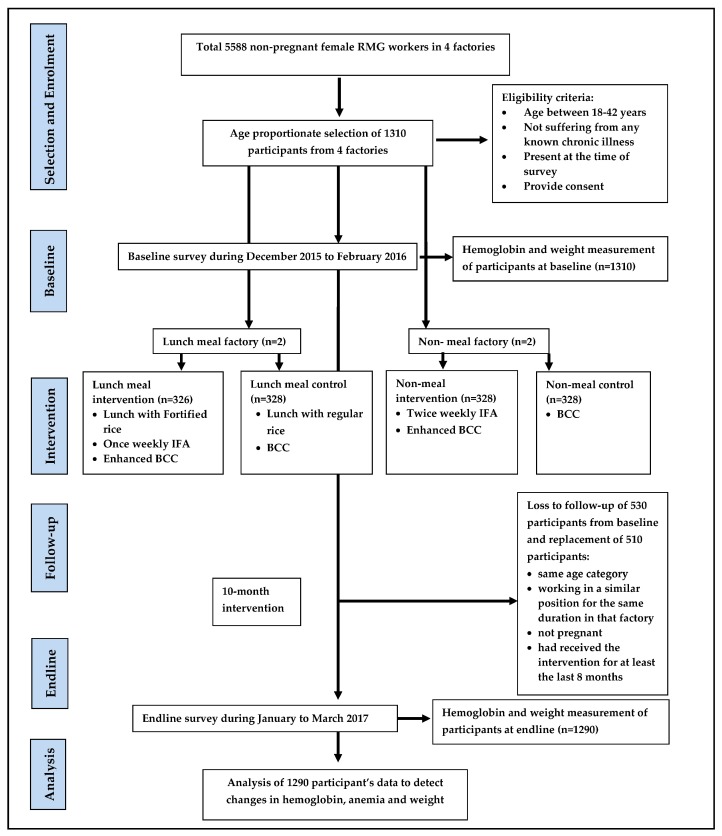
A flow chart of the study design.

**Figure 2 nutrients-11-01259-f002:**
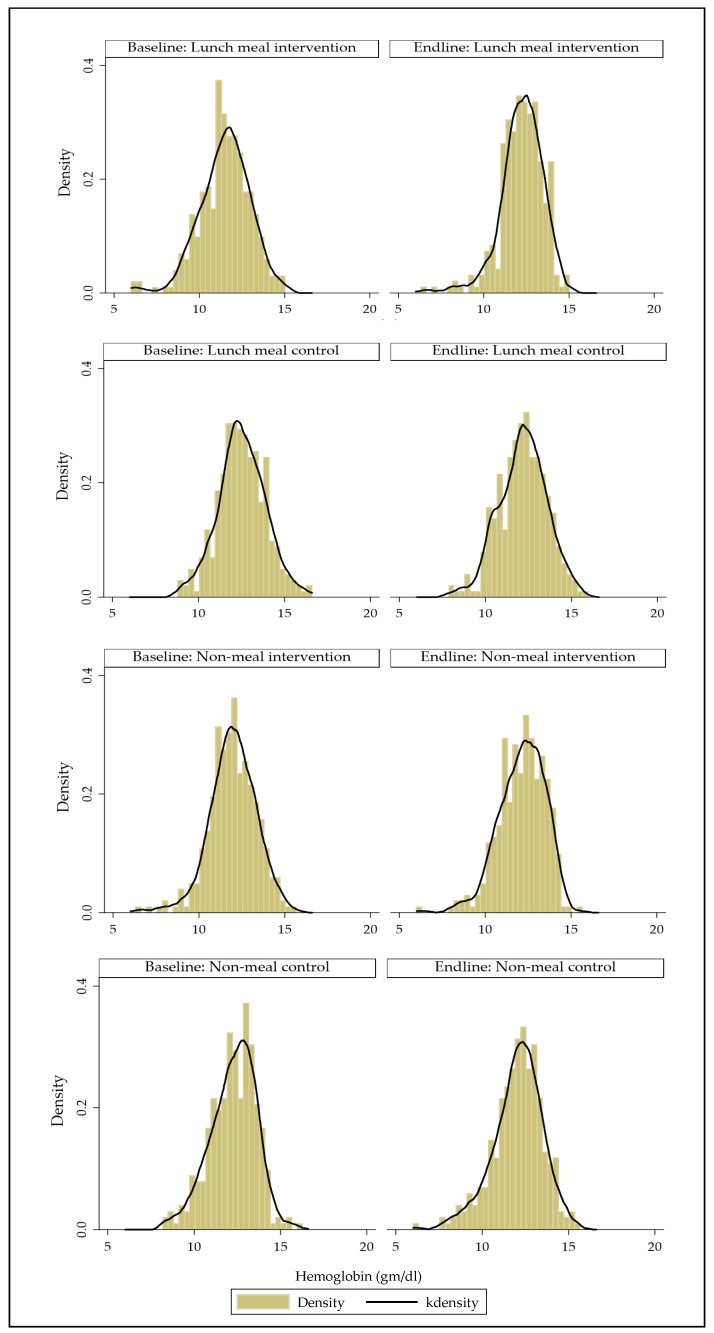
Distribution of hemoglobin at baseline and endline in each group (histogram with kernel density curve).

**Figure 3 nutrients-11-01259-f003:**
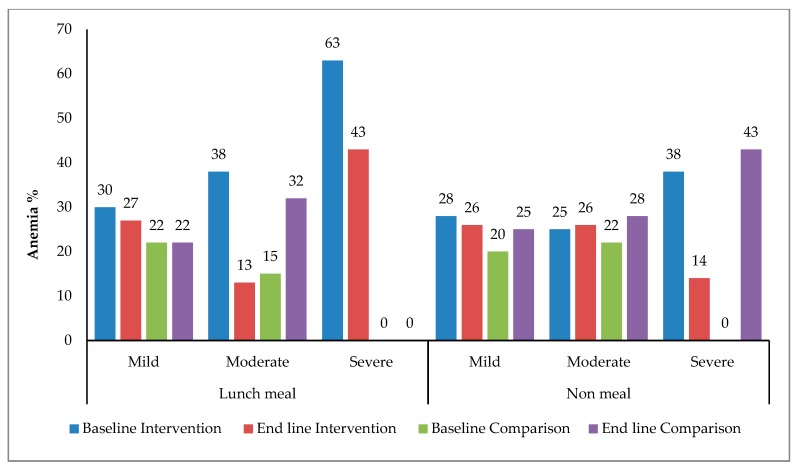
Changes in severity of anemia over time.

**Table 1 nutrients-11-01259-t001:** Summary of the package of interventions in the intervention and control factories with and without a lunch program.

**Package of Interventions for Factories with A Lunch Program**		
No. factories	1 Control factory (B)	1 Intervention factory (A)
Lunch	The factory provides the usual lunch meal with non-fortified rice and lentils and a small portion of single/mixed vegetable daily, meat or fish or egg three times weekly. The food was cooked with fortified oil and iodized salt. Rice and lentils were served daily in unlimited portions.	The factory provides a lunch meal enhanced with micronutrient fortified rice, as well as a diverse diet which includes animal sources foods 3 times per week, an egg at least one day per week, pulses and fortified rice and 1 larger portion of vegetable every day including a serving of green leafy vegetables 6 days per week. The food was cooked with fortified oil and iodized salt. Rice and lentils were served daily in unlimited portions. Lentils were made with twice the regular lunch lentil content per cooked volume.
Supplements	No IFA or other nutritional supplements	Once weekly IFA supplement to female factory workers; those reporting to be pregnant offered once daily supplements
BCC activities	Regular BCC modules include: Eating healthy Maternal health Reproductive health and family planning Sexually transmitted infections Malaria and Dengue Personal Hygiene Serious Illness Reproductive Cancer Waterborne Diseases Your body and menstruation	Enhanced BCC Module: regular BCC module in addition with: 1. Anemia 2. Nutrition and Dietary Diversity 3. Infant and Young Child Nutrition (IYCN) + Breastfeeding (BF)
Package of Interventions for factories without a lunch program		
No. factories	1 Control factory (D)	1 Intervention factory (C)
Supplements	No IFA or other nutritional supplements provided to workers through the factory	Factory provided twice weekly IFA supplement to female factory workers; those reporting to be pregnant was offered once daily supplements
BCCactivities	Regular BCC module (Same modules as described above)	Enhanced BCC Module (Same modules as described above)

Abbreviations: BCC: behavior change communication; IFA: iron-folic acid.

**Table 2 nutrients-11-01259-t002:** Baseline socio-demographic characteristics of the participants from four readymade garment factories.

Variables	Lunch Meal			Non-Meal		
	I (*n* = 326)	C (*n* = 328)	*p*	I (*n* = 328)	C (*n* = 328)	*p*
n (%)	Factory A	Factory B		Factory C	Factory D	
Age in years						
18–22	60 (18.4)	47 (14.3)	NS	40 (12.2)	29 (8.8)	NS
23–27	110 (33.7)	127 (38.7)		109 (33.2)	88 (26.8)	
28–32	100 (30.7)	101 (30.8)		109 (33.2)	129 (39.3)	
33–37	36 (11.0)	41 (12.5)		56 (17.1)	63 (19.2)	
38–42	20 (6.1)	12 (3.7)		14 (4.3)	19 (5.8)	
Worker type						
Fresher/trainee	14 (4)	12 (3.7)	NS	29 (8.8)	17 (5.2)	NS
Permanent	312 (95.7)	316 (96.3)		299 (91.2)	311 (94.8)	
Religion						
Islam	316 (96.9)	317 (96.6)	NS	291 (88.7)	306 (93.3)	NS
Hindu	10 (3.1)	10 (3.0)		35 (10.7)	22 (6.7)	
Christian	0 (0)	1 (0.3)		2 (0.6)	0 (0)	
Marital status						
Married	238 (73.0)	259 (79)	0.004	264 (80.5)	281 (85.7)	0.005
Unmarried	58 (17.8)	30 (9.1)		36 (10.9)	14 (4.3)	
Divorced/separated/widow	30 (9.2)	39 (11.9)		28 (8.5)	33 (10.1)	
Household ownership						
Own house	9 (2.8)	165 (50.3)	<0.001	8 (2.4)	14 (4.3)	NS
Rented house	317 (97.2)	163 (49.7)		320 (97.6)	314 (95.7)	
Education						
Formal education	286 (87.7)	297 (90.5)	NS	260 (79.3)	278 (84.8)	NS
Asset quintile						
Poorest	61 (19)	93 (28)	<0.001	83 (25.3)	81 (24.7)	NS
Poorer	58 (18)	86 (26)		91 (27.7)	71 (21.7)	
Middle	63 (19)	61 (19)		63 (19.2)	83 (25.3)	
Richer	59 (18)	58 (18)		62 (18.9)	57 (17.4)	
Richest	85 (26)	30 (9)		29 (8.8)	36 (11.0)	
Works at overtime, n (%)	316 (97)	323 (98)	0.018	309 (94)	325 (99)	0.001
Hours of overtime per months, mean (SD)	33 (10)	41 (9)	<0.001	44 (19)	47 (12)	0.02
Total Income in USD, median (IQR)	100 (87.5, 108.8)	108 (100, 112.7)	<0.001	104 (91.2, 117.7)	108.8 (97.5, 116.3)	0.006
Total Expenditure in USD, median (IQR)	93.8 (84.6, 106.3)	100 (87.5, 114.4)	0.001	101.3 (87.5, 112.5)	106.3 (82.5, 112.5)	NS
Anemia, n (%)	198 (60.7)	109 (33.2)	<0.001	157 (47.9)	119 (36.3)	0.002

Abbreviation: I: intervention; C: control; IQR: interquartile range; NS: *p*-values were not significant at 0.05 or 5% level; USD: US dollar. Continuous variables are presented by mean ± SD and categorical variables are presented as the percentage of participants (%). Note: 1 USD = 80.00 Bangladeshi Taka.

**Table 3 nutrients-11-01259-t003:** Changes in food and nutrition knowledge from baseline to endline.

Variables	Lunch Meal						Non-Meal					
	Baseline	Endline	*p*	Baseline	Endline	*p*	Baseline	Endline	*p*	Baseline	Endline	*p*
	I (326)	I (306)		C (328)	C (328)		I (328)	I (328)		C (328)	C (328)	
Knowledge on:	Factory A			Factory B			Factory C			Factory D		
Main food groups	27 (8.3)	178 (58.2)	<0.001	43 (13.1)	193 (58.8)	<0.001	26 (7.9)	151 (46)	<0.001	23 (7.0)	137 (41.8)	<0.001
Vitamins and minerals	52 (16.0)	202 (66)	<0.001	69 (21.0)	221 (67.4)	<0.001	41 (12.5)	147 (44.8)	<0.001	36 (11.0)	137 (41.8)	<0.001
Benefits of vitamin A containing foods	282 (86.5)	299 (97.7)	<0.001	254 (77.4)	312 (95.1)	<0.001	243 (74.1)	308 (93.9)	<0.001	234 (71.3)	311 (94.8)	<0.001
Iron containing foods	63 (19.3)	232 (75.8)	<0.001	62 (18.9)	184 (56.1)	<0.001	39 (11.9)	181 (55.2)	<0.001	48 (14.6)	146 (44.5)	<0.001
Benefits of Iron containing foods	245 (75.2)	295 (96.4)	<0.001	236 (71.9)	308 (93.9)	<0.001	218 (66.5)	297 (90.6)	<0.001	220 (67.1)	289 (88.1)	<0.001
Availability of Vitamin A fortified oil	55 (16.8)	133 (43.5)	<0.001	85 (25.9)	162 (49.4)	<0.001	68 (20.7)	135 (41.2)	<0.001	58 (17.7)	137 (41.8)	<0.001

Abbreviation: I: intervention; C: control. Continuous variables are presented by mean ± SD and categorical variables are presented as the percentage of participants (%).

**Table 4 nutrients-11-01259-t004:** Changes in water, sanitation, and personal hygiene practices, self-reported sickness, and workplace absenteeism from baseline to endline.

Variables	Lunch Meal						Non-Meal					
	Baseline	Endline	*p*	Baseline	Endline	*p*	Baseline	Endline	*p*	Baseline	Endline	*p*
	I (326)	I (306)		C (328)	C (328)		I (328)	I (328)		C (328)	C (328)	
	Factory A			Factory B			Factory C			Factory D		
Availability of clean and safe drinking water at workplace	301 (92.3)	280 (91.5)	<0.001	320 (97.6)	327 (99.7)	0.03	311 (94.8)	322 (98.2)	0.03	311 (94.8)	319 (97.3)	NS
Reported hand washing after defecation	311 (95.4)	299 (97.7)	NS	328 (100)	326 (99.4)	NS	327 (99.7)	325 (99.1)	NS	311 (94.8)	322 (98.2)	0.03
Products used for menstrual hygiene management, n (%)												
Sanitary pad	88 (26.9)	136 (44.4)	<0.001	135 (41.2)	293 (46.2)	NS	98 (29.9)	151 (46)	<0.001	66 (20.1)	103 (31.4)	0.006
Cloth	231 (70.9)	159 (52)		184 (56.1)	317 (50)		210 (64.0)	160 (48.8)		233 (71.0)	199 (60.7)	
Factory rags	3 (0.9)	5 (1.6)		7 (2.1)	15 (2.4)		8 (2.4)	6 (1.8)		11 (3.4)	6 (1.8)	
Self-reported sickness in last 15 days, n (%)												
Diarrhea	12 (3.7)	10 (3.3)	NS	8 (2.4)	16 (2.5)	NS	6 (1.8)	8 (2.4)	NS	5 (1.5)	15 (4.6)	0.02
Dysentery	5 (1.5)	4 (1.3)	NS	2 (0.6)	4 (0.6)	NS	1 (0.3)	1 (0.3)	NS	6 (1.8)	2 (0.6)	NS
Fever	54 (16.6)	36 (11.8)	NS	70 (21.3)	69 (10.9)	<0.001	66 (20.1)	67 (20.4)	NS	51 (15.6)	55 (16.8)	NS
Common cold	104 (31.9)	70 (22.9)	<0.001	84 (25.6)	128 (20.2)	<0.001	88 (26.8)	83 (25.3)	NS	67 (20.4)	78 (23.8)	NS
Urinary tract infection	24 (7.4)	8 (2.6)	<0.001	18 (5.5)	15 (2.4)	<0.04	16 (4.9)	19 (5.8)	NS	11 (3.4)	12 (3.7)	NS
Joint pain	67 (20.6)	43 (14.1)	0.03	74 (22.6)	84 (13.3)	<0.001	82 (25)	87 (26.5)	NS	60 (18.3)	59 (18)	NS
Workplace absenteeism (in last 30 days preceding interview), n (%)												
Absence due to sickness	22 (6.8)	11 (3.6)	NS	24 (7.3)	21 (6.5)	NS	32 (9.8)	32 (9.8)	NS	13 (3.9)	24 (7.3)	0.02
Days of sickness absenteeism, median (IQR)	2 (1, 4)	3 (2, 4)	NS	2 (1, 4)	3 (2, 4)	NS	2 (1, 4)	3 (2, 5)	NS	2 (1, 4)	3 (2, 4)	NS

Abbreviation: I: intervention; C: control; NS: *p*-values were not significant at 0.05 or 5% level. Continuous variables are presented by mean ± SD and categorical variables are presented as the percentage of participants (%).

**Table 5 nutrients-11-01259-t005:** Changes in hemoglobin (gm/dL) and weight (kg) over time.

Variables	Lunch meAl						Non-Meal					
	Baseline	Endline	*p*	Baseline	Endline	*p*	Baseline	Endline	*p*	Baseline	Endline	*p*
	I (326)	I (306)		C (328)	C (328)		I (328)	I (328)		C (328)	C (328)	
	Factory A			Factory B			Factory C			Factory D		
Hemoglobin (gm/dL), mean (SD)	11.5 (1.5)	12.2 (1.2)	<0.001	12.5 (1.4)	12.1 (1.3)	0.001	11.9 (1.3)	12.1 (1.3)	NS	12.3 (1.3)	12.0 (1.4)	0.04
Weight in kg, mean (SD)	52.2 (8.8)	53.5 (8.9)	NS	51.3 (8.1)	53.1 (8.5)	<0.01	52.0 (9.3)	52.9 (9.5)	NS	52.8 (8.6)	54.3 (8.8)	0.03

Abbreviation: I: intervention; C: control; NS: *p*-values were not significant at 0.05 or 5% level. Continuous variables are presented by mean ± SD and categorical variables are presented as the percentage of participants (%).

**Table 6 nutrients-11-01259-t006:** Effect of intervention on anemia (% of women with any anemia) and hemoglobin concentration (gm/dL): DID analysis.

Indicator		I before (i)	I after (ii)	Difference in I (ii–i)	C before (iii)	C after (iv)	Differece in C (iv–iii)	Baseline difference, (B = i–iii)	Endline difference, (E = ii–iv)	Unadjusted difference-in-difference (DID) (E–B)	Adjusted DID
**Anemia (%)**											
Model 1	Lunch meal (A vs. B)	60.7	36.9	−23.8 **	33.2	41.2	8.0	27.5 **	−4.3	−31.8 **	−32.4 **
Model 2	Non-meal (C vs. D)	47.9	41.5	−5.5	36.3	41.8	6.4	11.6	−0.3	−11.9 *	−11.6b *
**Hemoglobim (gm/dL)**											
Model 1	Lunch meal (A vs. B)	11.50	12.23	0.73	12.53	12.17	−0.36	−1.03 **	0.05	1.08 **	1.05 **
Model 2	Non-meal (C vs. D)	11.98	12.14	0.16	12.27	12.05	−0.22	−0.29 **	0.10	0.39 **	0.40 **

Abbreviation: I: intervention; C: control. Indicator variable anemia changes are presented as the percentage points (%) and hemoglobin concentration changes are presented as mean (gm/dL). Model 1: adjusted for marital status, asset index, household ownership, overtime work hours per month, baseline anemia difference and intracluster correlation (ICC); Model 2: adjusted for marital status, overtime work hours per month and intracluster correlation (ICC). * *p* < 0.01; ** *p* < 0.001.

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
