# Peer review of "Effectiveness of Workplace Nutrition Programs on Anemia Status among Female Readymade Garment Workers in Bangladesh: A Program Evaluation"

_nutrients, 2019, doi:10.3390/nu11061259_

Reviewer 1 Report

Major comments

1.   The title is too dogmatic. A small study in 4 factories will add to the literature, but does not conclusively demonstrate a causal effect.

2.   The abstract is confusing. It is difficult to work out what the active intervention is in the two pairs of factories.

3.   The duration of the intervention needs to be in the abstract.

4.   The discussion does not mention the possibility of confounding, ie something external happened to the intervention factories and their communities to decrease anemia, eg community screening program.

5.   Close examination of the mean hemoglobins (table 6) and changes in anemia prevalence generate questions that are not discussed in the discussion:

A.   The lunch meal factories have big differences in the anemia prevalence. They are clearly different communities, and any intervention that worked would produce a bigger impact on the factory with the higher prevalence of anemia if it was applied to both factories (as there is much more capacity for improvement).

B.   Table 6 provides a comparison of before and after haemoglobin levels, which is an alternative analysis which has more power than the following binary anemia analysis (table 7). This is informative as haemoglobin decreased in both control factories, so the apparent improvement could be due to a third factor that led to more anemia in the control factories. There was no significant change in mean haemoglobin in Factory C, which is odd if the intervention was deemed to be effective.  

Minor comments

1.   What was the response rate of participants?

2.   P2, L73. ‘ultimately reduces anemia’. The syntax/grammar needs review, and a reference to support this statement needs to be placed.

3.   P13, L424. ‘all sort’.

Author Response

Response to Reviewer 1 Comments

 Major comments

1.   The title is too dogmatic. A small study in 4 factories will add to the literature, but does not conclusively demonstrate a causal effect.

Response 1: Thank you for your valuable suggestion. We agree with your point and revised the title.

2.   The abstract is confusing. It is difficult to work out what the active intervention is in the two pairs of factories.

Response 2: We have provided additional information in the abstract to clear the confusion.

3.   The duration of the intervention needs to be in the abstract.

Response 3: The duration of the intervention was 10 months and it is already mentioned in the abstract.

4.   The discussion does not mention the possibility of confounding, ie something external happened to the intervention factories and their communities to decrease anemia, eg community screening program.

Response 4: Thank you for the query. We have asked the participants about other programs or supplementation during the qualitative in-depth interview and we did not find any. We now have added this point in the methods, results section and discussion.

5.   Close examination of the mean hemoglobins (table 6) and changes in anemia prevalence generate questions that are not discussed in the discussion:

A.   The lunch meal factories have big differences in the anemia prevalence. They are clearly different communities, and any intervention that worked would produce a bigger impact on the factory with the higher prevalence of anemia if it was applied to both factories (as there is much more capacity for improvement).

Response 5A: We agree with your point. We have mentioned about the big difference in baseline anemia among lunch meal intervention factories in discussion under study limitation. We have tried to overcome this limitation by adjusting the baseline differences in the model. We also mentioned in limitation that, “Although, we have a speculation that the impact of intervention might have been different if the comparison group also had a higher prevalence of anemia.”

B.   Table 6 provides a comparison of before and after haemoglobin levels, which is an alternative analysis which has more power than the following binary anemia analysis (table 7). This is informative as haemoglobin decreased in both control factories, so the apparent improvement could be due to a third factor that led to more anemia in the control factories. There was no significant change in mean haemoglobin in Factory C, which is odd if the intervention was deemed to be effective.

Response 5B: Thank you for your close observation. As a response to the reviewer 2 comments, we analyzed the effect of the intervention on haemoglobin (Hb) concentration by calculating DID (table 8) and found similar results as table 6 but the power was much higher. The factory C has slight changes in Hb concentration in the binary model which is statistically insignificant but even this slight improvement in mean concentration is quite important in countries like Bangladesh where anemia is so prevalent among RMG workers. Considering the main outcome as anemia, the intervention can be considered effective.

From IDI’s we identified a lack of support during sickness and family emergency and salary cut during unauthorized leave in control factories. This could be the third factor that might have caused an increase in anemia from baseline due to lack of support during recovery. In addition, the enhanced BCC in intervention factories might helped workers to practice appropriately during their home stay which might stop the continuous progression of anemia for intervention factory, which did not happen for the control group. We now have added this in the discussion section.

 Minor comments

1.   What was the response rate of participants?

Response 1: The response rate of the participants was 100%. We now have mentioned about this in the methods and results sections.

2.   P2, L73. ‘ultimately reduces anemia’. The syntax/grammar needs review, and a reference to support this statement needs to be placed.

Response 2: Thank you for the suggestion. We have revised the line and added a reference. Please see P2, L82-83.

3.   P13, L424. ‘all sort’.

Response 3: We now have removed the term ‘all sort’. Please see P16, L544.

Reviewer 2 Report

This manuscript aims to present the results from the impact evaluation of a very original program, aiming at testing different intervention packages based on daily nutritionally improved lunch and IFA tablets and behavior change counseling, delivered in Readymade Garment factories to decreased anemia in workers - women of reproductive age. The platform used for delivering nutritional products and nutritious foods presented here is very interesting and of great public health importance. However, some weaknesses in the study design would deserve more attention and further explanation.

1.       Introduction/description of the intervention:

The paper would benefit from providing more information on fortified rice and IFA tablets, as the main outcome of interest is anemia.

This could be presented in the method section. For example, what was the dosage of iron and folic acid in the IFA tablets delivered to women? What kind of iron was included in the premix used in the fortified rice. Was it EDTA iron? Fumarate? One would expect that the total amount of iron delivered through fortified rice (6 mg) AND IFA would be discussed later in the discussion section, as well as iron bioavailibity regarding the rest of the diet.

2.       Study design:

a.       The study design does not bring high level of evidence (not state of the art design). As mentioned in the discussion, the authors are aware of the weakness of their design, such as not having randomized at cluster or individual level into treatments groups (or interventions groups). Here, what is not clear is the unit of analysis which seems the factory. There are 2 paired intervention-controls groups and only 4 factories. Please explain.

b.       The research questions that comparisons are aimed to answer are not clear. With this design, authors cannot answer which component is having impact. What is the additional benefit of adding IFA? What was the purpose of adding IFA? Was there an assumption that the lunches would be insufficient to decrease anemia? What is the additional benefit of BCC as it was delivered in all groups? Please explain the rationale behind the choice of the different components in the study groups.

3.       Main outcome: The prevalence of anemia is the main outcome, but why not presenting also the impact of the interventions on hemoglobin concentration (DID)?

4.       Statistical analysis: It’s a quasi-experimental design (no randomization). It is not clear what is the unit of analysis, with the statistical methods (DID) used. Individual or cluster level? Please explain why taking into account an ICC? Assessing changes in anemia and hemoglobin levels between groups could be done using mixed-effects regression models with random effects at the individual level (repeated measures).

5.       Outcomes and indicators

a.       Knowledge indicators are coded from qualitative interviews, then used as quantitative data (to run statistical tests of comparisons between groups). This is not common practice and state of the art. Data collected through qualitative interviews and coded in a second phase as quantitative data can be of poor quality (without minimum controls and skip patterns, as included in most computer assisted program interviews). Qualitative data in mixed methods approach is rather used to support quantitative results (triangulate), to bring in-depth quality information.

b.       In addition, there is no information on how the indicators were constructed (how many items to calculate each indicator, cutoffs of responses, etc…)

6.       Drop out: The drop out is important (50 to 60%). The first thing would be to test if the workers who had left at endline had different characteristics than those who stayed. Then, a propensity score could be calculated and included as a control covariate in the regression model. This could have been a different option than replacement of interviewees at endline.

Author Response

Response to Reviewer 2 Comments

Comments and Suggestions for Authors

This manuscript aims to present the results from the impact evaluation of a very original program, aiming at testing different intervention packages based on daily nutritionally improved lunch and IFA tablets and behavior change counseling, delivered in Readymade Garment factories to decreased anemia in workers - women of reproductive age. The platform used for delivering nutritional products and nutritious foods presented here is very interesting and of great public health importance. However, some weaknesses in the study design would deserve more attention and further explanation.

1.      Introduction/description of the intervention:

The paper would benefit from providing more information on fortified rice and IFA tablets, as the main outcome of interest is anemia.

This could be presented in the method section. For example, what was the dosage of iron and folic acid in the IFA tablets delivered to women? What kind of iron was included in the premix used in the fortified rice. Was it EDTA iron? Fumarate? One would expect that the total amount of iron delivered through fortified rice (6 mg) AND IFA would be discussed later in the discussion section, as well as iron bioavailability regarding the rest of the diet.

Response 1: Thank you for these important questions. We have added the information’s on fortified rice and IFA tablets in the methods section. We also have added the information on the total amount of iron delivered through fortified rice and IFA in the discussion section. We added the information in details in supplementary files.

2.      Study design:

a.      The study design does not bring high level of evidence (not state of the art design). As mentioned in the discussion, the authors are aware of the weakness of their design, such as not having randomized at cluster or individual level into treatments groups (or interventions groups). Here, what is not clear is the unit of analysis which seems the factory. There are 2 paired intervention-controls groups and only 4 factories. Please explain.

Response 2a: Thank you for the query. The unit of analysis is the factory. There are two types of intervention package tested against two consecutive controls; so there are two paired intervention-controls groups, which equals to four factories. Among factories that already provide lunch to workers, one intervention (A) and one control (B) factory were selected, and among factories that do not provide lunches to their workers, one intervention (C) and one control (D) factory were selected. We already mentioned it in the methods section. Now, we have included this in our abstract also.

b.      The research questions that comparisons are aimed to answer are not clear. With this design, authors cannot answer which component is having impact. What is the additional benefit of adding IFA? What was the purpose of adding IFA? Was there an assumption that the lunches would be insufficient to decrease anemia? What is the additional benefit of BCC as it was delivered in all groups? Please explain the rationale behind the choice of the different components in the study groups.

Response 2b: We have now added the rationale for each intervention component on the methods section in details. We also added additional tables as supporting documents. Please see Table S1, Table S2, and Table S3.

3.      Main outcome: The prevalence of anemia is the main outcome, but why not presenting also the impact of the interventions on hemoglobin concentration (DID)?

Response 3: Thanks for the suggestion. We now have added the impact of the intervention on hemoglobin concentration (DID) as a separate table 8 and added the information in abstract, results and discussion section.

4.      Statistical analysis: It’s a quasi-experimental design (no randomization). It is not clear what is the unit of analysis, with the statistical methods (DID) used. Individual or cluster level? Please explain why taking into account an ICC? Assessing changes in anemia and hemoglobin levels between groups could be done using mixed-effects regression models with random effects at the individual level (repeated measures).

Response 4: In our analysis, each factory was our cluster. Moreover, within each factory, the characteristics of the individuals (factory workers) were correlated. Therefore, we adjust the intra-cluster correlation (ICC) in mixed effect linear model at the cluster level (factory). Since we adjusted for TIME (repeated measures) variable in the mixed effect model, we did not use random effects at the individual level; as both will serve a similar purpose.

However, as per your suggestion, we also did a mixed-effect model with a random effect at the individual level and find the same results.

5.      Outcomes and indicators

a.       Knowledge indicators are coded from qualitative interviews, then used as quantitative data (to run statistical tests of comparisons between groups). This is not common practice and state of the art. Data collected through qualitative interviews and coded in a second phase as quantitative data can be of poor quality (without minimum controls and skip patterns, as included in most computer-assisted program interviews). Qualitative data in mixed methods approach is rather used to support quantitative results (triangulate), to bring in-depth quality information.

Response 5a: Our sincere apologies for the confusion. The knowledge indicators that were presented in table 3 were quantitative variables collected during baseline and endline surveys. We then further triangulated the findings with qualitative in-depth interviews. The statistical tests were run based on the responses from quantitative surveys. We did not run statistical analysis based on the qualitative responses. We revised our abstract and result section to clear this confusion.

b.      In addition, there is no information on how the indicators were constructed (how many items to calculate each indicator, cutoffs of responses, etc…)

Response 5b: The results reported in table 3 are all quantitative responses based on single questions to indicators measure each indicator. The response rates were 100% for all qualitative and quantitative responses. We added about this in the methods and results section.

6.      Drop out: The drop out is important (50 to 60%). The first thing would be to test if the workers who had left at endline had different characteristics than those who stayed. Then, a propensity score could be calculated and included as a control covariate in the regression model. This could have been a different option than replacement of interviewees at endline.

Response  6: We really appreciate for your suggestion. Our overall study dropout rate was between 30-40%. We checked the characteristics of workers who left at endline and who stayed, we did not find any significant differences in the socio-demographic characteristics as well as the outcome variable, which is anemia. We also agree that, calculating the propensity score could have been a good option. We were not aware of this method previously and thus we used the replacement method at endline.
